# *Mycobacterium abscessus* Genetic Determinants Associated with the Intrinsic Resistance to Antibiotics

**DOI:** 10.3390/microorganisms9122527

**Published:** 2021-12-07

**Authors:** Mylene Gorzynski, Tiana Week, Tiana Jaramillo, Elizaveta Dzalamidze, Lia Danelishvili

**Affiliations:** 1Department of Biomedical Sciences, Carlson College of Veterinary Medicine, Oregon State University, Corvallis, OR 97331, USA; gorzynsm@oregonstate.edu (M.G.); tslweek@gmail.com (T.W.); jaramilt@oregonstate.edu (T.J.); bliznyue@oregonstate.edu (E.D.); 2Department of Biochemistry & Molecular Biology, Oregon State University, Corvallis, OR 97331, USA; 3Department of Bioengineering, College of Engineering, Oregon State University, Corvallis, OR 97331, USA; 4Department of Animal Sciences, College of Agricultural Sciences, Oregon State University, Corvallis, OR 97331, USA; 5BioHealth Sciences, Department of Microbiology, College of Sciences, Oregon State University, Corvallis, OR 97331, USA

**Keywords:** *M. abscessus*, rapidly growing mycobacteria, intrinsic resistance, virulence factors, efflux pumps, biofilm, macrophage, antibiotic treatment

## Abstract

*
Mycobacterium abscessus
*
subsp. *abscessus* (MAB) is a fast-growing nontuberculous mycobacterium causing pulmonary infections in immunocompromised and immunocompetent individuals. The treatment of MAB infections in clinics is extremely challenging, as this organism is naturally resistant to most available antibiotics. There is limited knowledge on the mechanisms of MAB intrinsic resistance and on the genes that are involved in the tolerance to antimicrobials. To identify the MAB genetic factors, including the components of the cell surface transport systems related to the efflux pumps, major known elements contributing to antibiotic resistance, we screened the MAB transposon library of 2000 gene knockout mutants. The library was exposed at either minimal inhibitory (MIC) or bactericidal concentrations (BC) of amikacin, clarithromycin, or cefoxitin, and MAB susceptibility was determined through the optical density. The 98 susceptible and 36 resistant mutants that exhibited sensitivity below the MIC and resistance to BC, respectively, to all three drugs were sequenced, and 16 mutants were found to belong to surface transport systems, such as the efflux pumps, porins, and carrier membrane enzymes associated with different types of molecule transport. To establish the relevance of the identified transport systems to antibiotic tolerance, the gene expression levels of the export related genes were evaluated in nine MAB clinical isolates in the presence or absence of antibiotics. The selected mutants were also evaluated for their ability to form biofilms and for their intracellular survival in human macrophages. In this study, we identified numerous MAB genes that play an important role in the intrinsic mechanisms to antimicrobials and further demonstrated that, by targeting components of the drug efflux system, we can significantly increase the efficacy of the current antibiotics.

## 1. Introduction

*Mycobacterium abscessus* subsp. *abscessus* (MAB) is a fast-growing non-tuberculous mycobacterium (NTM) responsible for severe respiratory and mucosal infections in patients with preexisting lung conditions such as chronic obstructive pulmonary disease, cystic fibrosis, and emphysema [1,2]. In the United States, pulmonary *M. abscessus* complex infections range 2.6–13.0% [3,4], with successful treatment rates in only 45.6% of patients, highlighting a need for more satisfactory therapy strategies [5]. An alarming rise of MAB incidence in recent years has been related to the increased number of immunocompromised patients and, in part, the global population aging [6,7].

MAB lung infections are very difficult to treat due to their natural (intrinsic) resistance to most available antibiotics [4]. While the treatment regimens are extended for twelve months or longer, they lack complete efficiency in clinics. The current recommended treatment regiments for MAB consist of intravenous and multidrug therapy with macrolide and aminoglycoside or beta-lactams [8]. The common antibiotics used are amikacin (AMK) of the aminoglycoside functional class, azithromycin or clarithromycin (CLA) of the macrolide class, and cefoxitin (FOX) or imipenem from the beta-lactam class [8,9]. Recent studies have identified MAB strains with genetic polymorphisms of target genes that confer resistance to specific antibiotics [10,11], further complicating the treatment outcome. For example, about 20% of MAB isolates in US have specific point mutations in the *erm*(41) gene, conferring resistance to CLA [12]. It has been shown that *M. abscessus* subsp. *massiliense* has a truncated *erm*(41) gene when compared with *M. abscessus* subsp. *abscessus,* explaining the different prevalence of macrolide resistance in different subspecies [4,13]. In addition, while the resistance to aminoglycosides is mainly associated with mutations in the *rrs* gene (encoding the 16S rRNA) or aminoglycoside-modifying enzymes, mutations in the *rrl* gene that encode the 23S rRNA have been shown to cause macrolide resistance [9,14].

Furthermore, mycobacterial highly hydrophobic cell walls that are comprised of crosslinked peptidoglycan, branched arabinose, and mycolic acids [15,16] represent one of the major permeability barriers to hydrophilic antibiotics in entering cells. The inhibition of lipid transport machinery for cell wall assembly and maintenance, such as lipoprotein glycosylation, has been demonstrated to result in a greater susceptibility to antibiotics [17]. The cell surface transport systems of MAB, such as influx and efflux machineries, have been shown to represent the major contributing factors of intrinsic resistance to aminoglycosides, macrolides, and quinolones [5,12,18,19]. The channel-forming outer membrane porins embedded in the cell wall allow for the entry of substrates and small and hydrophilic antibiotics into the cell while also playing an important role in mycobacterial survival and virulence [20]. In the presence of antibiotics, bacteria can downregulate porins to avoid uptake and killing by antimicrobials. Efflux pumps, on the other hand, aid bacteria in pumping out toxins and antibiotics before they can significantly impact the cell. A reduction in efflux pump activity using inhibitor compounds has been shown to decrease the resistance to antibiotics [9,21]. Therefore, these transport mechanisms have been utilized as targets for antimycobacterial drugs [16].

Other factors of mycobacterial intrinsic resistance include antibiotic inactivating enzymes that can also specifically target certain antibiotics. For example, MAB contains the beta-lactamase Bla_Mab_ that hydrolase beta-lactams, making them ineffective antibiotics for treatment options [22]. Beta-lactamase inhibitors such as avibactam have been shown to effectively inhibit Bla_Mab_ and restore MAB susceptibility to beta-lactams [22]. Another enzyme, rifampicin ADP-ribosyl transferase ArrMAB (*MAB_0591*), has been demonstrated to inactivate rifamycin antibiotics such as rifabutin and rifampin [23]. MAB also possess monooxygenase enzymes that contribute to rifampicin inactivation [24]. In addition, MAB adaptabilities and phenotypic remodeling that take place within the host are major challenges of finding effective treatment solutions for MAB. The global analysis studies have shown that the cell wall modifications and metabolic changes that MAB evolves within the intracellular environment during the host infection aids the pathogen to resist the bactericidal concentrations of antimicrobials [25].

Many MAB genetic elements related to bacterial intrinsic resistance that subsequently contribute to the development of antibiotic resistance are unknown, and the discovery of these factors is a fundamental gap to fill in the field. In attempts to identify MAB genes that are linked to antibiotic intrinsic resistance, we screened the MAB transposon library and found gene knockout mutants that were either more susceptible or resistant to antibiotics when compared with the wild-type strain. The selected clones were further characterized and validated for an increased efficacy of the clinically available antibiotics in cultured macrophages. The new knowledge obtained in this study can be used for the development of new and more effective strategies that can overcome MAB intrinsic resistance while increasing the potency of the current therapy regimens.

## 2. Materials and Methods

### 2.1. Mycobacterial Strains and Culture Conditions

The reference-type strain *M. abscessus* subsp. *abscessus* 19977 (hereafter, MAB 19977 or the same as the wild-type MAB 19977) with a smooth colony phenotype was purchased from the American Type Culture Collection (ATCC, Manassas, VA, USA). MAB clinical isolates were obtained in collaboration with the Cystic Fibrosis Research and Development Program at the National Jewish Health in Denver, CO, USA. The isolates used included the NR49093 strain DJO44274, NR44273 strain 4529, NR442746 strain 4530, DNA00703, DNA01163, DNA01627, DNA01639, and DNA01715. All strains were preserved at −80 °C until use. The MAB strains were cultured on 7H10 Middlebrook agar or 7H9 Middlebrook broth (Hardy Diagnostics, Santa Maria, CA, USA) supplemented with 5% oleic acid and albumin dextrose and catalase (OADC, Hardy Diagnostics, Santa Maria, CA, USA). The mid-log phase-grown bacteria (3–5 days) were used for inoculum preparations in Hank’s Balanced Salt Solution (HBSS; VWR, Visalia, CA, USA) and were adjusted to the McFarland Standard 1.0 (approximately 3 × 10^8^ bacteria/mL). Bacterial serial dilutions were plated on 7H10 agar for the colony-forming unit (CFU) counts to record the exact concentrations of MAB inoculum used in the experiments.

### 2.2. Creation of MAB Mutant Library Using MycomarT7 Transduction and Transposon Mutagenesis

The MAB 19977 gene knockout library was created using MycomarT7 (Mmt7), a temperature-sensitive transposon-containing phagemid provided by Eric Rubin at the Harvard T.H. Chan School of Public Health, Boston, MA, USA. The procedures for the generation of the Mmt7 mutant library were previously described [26]. Briefly, MAB 19977 was grown until the mid-log phase at 37 °C in 7H9 broth supplemented with 10% OADC and 0.1% Tween-80 with shaking at 200 rpm. Bacteria were washed two times with a mycobacteriophage (MP) buffer (150-mM NaCl, 50-mM (pH 7.5) Tris-HCl, 10-mM MgSO_4_, and 2-mM CaCl_2_) and diluted to a 3 × 10^8^ CFU/mL MP buffer using the McFarland Standard 1.0. Each milliliter of MAB suspension was infected with Mmt7 at a multiplicity of infection (MOI) of two and transduced at 37 °C for 4 h with intermittent mixing. The aliquots were plated on 7H10 agar containing 50-μg/mL kanamycin, and individual kanamycin-resistant colonies were further tested with PCR for the presence of Mmt7 transposon. Out of one hundred colonies tested, all the mutants were positive for Mmt7. The library was generated by storing individual colonies in 96-well plates.

### 2.3. The High-Throughput Screening of the MAB Mutant Library for Antibiotic Susceptibility

The primary screening for antibiotic susceptibility was performed by utilizing a library of 2000 MAB gene knockout mutants and three antibiotics: clarithromycin (CLA, TCI AMERICA, Portland, OR, USA), amikacin (AMK; Sigma-Aldrich, St. Louis, MO, USA), and cefoxitin (FOX; Sigma-Aldrich, St. Louis, MO, USA) at the minimal inhibitory concentration (MIC) and the bactericidal concentration (BC), listed in Table 1. Seven replicate plates of the library were cultured in 7H9 Middlebrook broth in 96-well plates and at an optical density (OD) of 0.05. One replicate plate without any antibiotic addition served as a mutant growth control, while the other six copies were exposed to MIC and BC concentrations of AMK, CLA, and FOX and incubated in a shaker at 37 °C for 5 days. The optical density (OD_600_) for each plate was measured on an Epoch microplate spectrophotometer (BioTek, Winooski, VT, USA). The mutant library growth in the antibiotic exposed plates was normalized from a growth control plate with no antibiotic exposure. In addition, the wild-type MAB 19977 served as a control for the identified transposon mutants that displayed a decreased susceptibility (no bacterial growth) to the MIC and increased resistance (bacterial growth) to the BC concentrations of antibiotics. Sensitive and resistant mutants from the parental strains were selected and subjected to a secondary confirmational assay by repeating the experiment three times. MAB clones with decreased susceptibility were termed as MIC mutants and mutants with increased resistance as BC mutants.

### 2.4. Ligation Mediated PCR (LM-PCR) for Analysis of MAB Gene Knockout Mutants

The selected MIC and BC mutants were sequenced as previously described [26]. Briefly, bacterial 1-mL turbid suspensions were created in dH_2_O, and 200 μL of 0.1-mm glass beads were added to the tubes. Bacteria were mechanically disrupted using the bead-beating (OMNI Bead Ruptor Elite) on a high speed for 30 s repeated 4 times and, between, placed on ice. The tubes were micro-centrifuged at 15,000 rpm for 1 min and processed for DNA extraction using the Zymo Research Clean and Concentrate DNA extraction kit (Zymo Research, Irvine, CA, USA) according to the manufacturer’s protocols. The DNA was quantified on the TECAN nanoquant machine and subjected to a single digestion with SalI or double digestion with BamHI and BglII (Thermo Fisher fast-digest enzymes) at a concentration range of 100–150 ng and for 20 min at 37 °C. Adapter oligos were created by combining equal molar amounts of each oligonucleotide for SalI (Salgd + Salpt adapter [26]) and BamHI/BglII (Salgd + Bampt adapter [26]) and with 1× Taq DNase buffer supplemented with MgCl_2._ The reaction was annealed by a gradual temperature reduction in a thermal cycler from 80 °C down to 4 °C over 1 h. Next, 25 ng of digested DNA was ligated with 0.5 μL of freshly prepared 100-µM adapters using T4 ligase for 1 h, followed by 10 min at 65 °C to inactivate the ligase. Two LM-PCR reactions were carried out using a Fidelitaq system (Affymetrix, Santa Clara, CA, USA) and 0.5-μL ligation mix with two primer sets: the first primer set of pmyco/pSalg and the second mmts7seqF2/pSalg [26]. Two sets of primers were used for each sample that shared the reverse primer (complementary to the adapter) and had the forward primer at either the inverted repeat of the transposon or 150 bp upstream into the transposon. The PCR reaction was set as the following: an initial denaturation at 95 °C for 5 min, 40 cycles of 95 °C for 30 s, 58 °C for 30 s, 72 °C for 1 min 45 s, and a final elongation at 72 °C for 10 min. The products were visualized on an agarose gel via electrophoresis, and amplicons that were larger than 150 bp were purified with the Thermo Fisher Scientific GeneJET Gel Extraction Kit according to the manufacture’s protocols (Thermo Fisher Scientific, Waltham, MA, USA). The mmt7seqF2 primer [26] was used for sample sequencing in the Center for Quantitative Life Sciences (CQLS) at Oregon State University, Corvallis, OR, USA.

### 2.5. Bioinformatic Analysis of DNA Sequencing

The sequenced mutants were analyzed against the reference genome of the *Mycobacterium abscessus* subsp. *abscessus* strain 19977 using the National Center for Biotechnology Information (NCBI) Basic Local Alignment Search Tool (BLAST). The gene domains/motifs were further analyzed using the NCBI conserved domain (CD-Search) and the Universal Protein Resource (UniProt) tools. The uncharacterized/conserved hypothetical gene sequences were also aligned to the well-characterized genome of the *M. tuberculosis* H37Rv strain to identify its homology, if any. In addition, the signal IP-5.0 server at the DTU Health Tech Center for Biological Sequence Analysis was used for information on the signal peptide location and cleavage sites.

### 2.6. The Antibiotic Susceptibility Testing with the Broth Microdilution Method

Sixteen MIC and BC mutants of MAB 19977 belonging to the transport systems were subjected to the third confirmational assay by reevaluating the antibiotic susceptibility with the broth microdilution method and establishing changed MIC/BC concentrations to AMK, CLA, and FOX by comparing them to the wild-type MAB 19977 as a control. The antibiotic susceptibility was performed in the range of 2–256 μg/μL for AMK, 0.125–64 μg/μL for CLA, and 4–256 μg/μL for FOX. The MAB mutants were grown until the mid-log phase on 7H10 Middlebrook agar plates, and the inoculums were prepared in HBSS by adjusting the bacterial turbidities to the McFarland Standard of 1.0 (3 × 10^8^ CFU/mL). The inoculums were further diluted to 3 × 10^6^ CFU/mL, and 100 μL was cultured into 3 mL of 7H9 Middlebrook broth supplemented with and without antibiotics to achieve a final 1 × 10^5^ CFU/mL concentration. The varying antibiotic MIC concentrations were obtained with the microdilution method. The wild-type MAB 19977 was included as a reference. The culture tubes were placed in a shaker incubator at 37 °C for 5 days and later analyzed based on a visual observation by comparing the tubes to nonantibiotic-exposed mutants and the wild type controls. Bacterial MICs were recorded as the lowest concentration (in μg/mL) of an antibiotic that inhibited the growth of a given strain/mutant of bacteria. In addition, tubes of the BC concentration were centrifuged at 10,000× *g* for 20 min and plated on 7H10 agar plates to confirm the absence of viable MAB.

### 2.7. RNA Extraction, cDNA Synthesis, and Real-Time Quantitative PCR

The 7H9 Middlebrook broth without antibiotic supplement or containing two times the MIC concentrations of AMK or CLA were prepared, and nine MAB clinical isolates (listed above), including MAB 19977, were cultured in 25 mL at a 1 × 10^8^ CFU/mL concentration. The flasks were agitated in the shaker for 8 h or 24 h at 37 °C, and the total RNAs were isolated using Direct-zol RNA miniprep, according to the manufacturer’s protocols. Briefly, the bacteria were centrifuged at 3500 rpm for 20 min and pellets resuspended in 600-μL TRIzol reagent (Invitrogen, Carlsbad, CA, USA), followed by rapid mechanical agitation in a bead beater (OMNI Bead Ruptor Elite) to lyse cells. The cellular debris was removed through centrifuging at 15,000× rpm for 2 min, and the supernatants were collected. The samples were mixed with 95% ethanol at a 1:1 ratio and then purified using the Direct-zol RNA miniprep kit according to the protocols provided by the manufacturer (Zymo Research, Irvine, CA, USA). Next, purified RNAa was treated with RNase-free DNase I (Fisher Scientific, Waltham, MA, USA) in the 10× DNase Roche Buffer for 1 h at 37 °C. The RNA concentrations were quantified using TECAN measuring at 260-nm absorbance, and the quality was determined with a 260/280-nm absorbance ratio. Ratios > 1.9 were accepted. The RNA samples were stored at −80 °C until required.

The cDNAs were synthesized using the iScript cDNA synthesis kit according to the manufacturer’s instructions (Bio-Rad, Hercules, CA, USA). The normalized RNAs to 200 ng/mL were treated with 1 μL of iScript reverse transcriptase and 4 μL of iScript reaction mix containing a blend of oligo(dT) and random hexamer primers. The samples were incubated in a thermocycler following the Bio-Rad iScript cDNA reaction protocols.

The RT-qPCR was performed in a CFX Connect Real-Time PCR machine (Bio-Rad, Hercules, CA, USA) using the IQ SYBR Green Supermix detection system. The PCR reactions were set in duplicates and in 10-μL reactions consisting of 5 μL of IQ SYBR Green Supermix, 1 μL of each primer (10 μM; Table 2), 1 μL of cDNA, and 2-μL molecular grade water and with the following PCR protocol: 95 °C for 5 min, 95 °C for 10 s, and 57 °C for 30 s for 39 cycles, ended with a melting curve at 65 °C for 5 s and 95 °C for 5 s. The relative mRNA expression levels were determined with the comparative quantification cycle (Cq) method based on cycle threshold (CT) values [27]. The relative expressions of 15 target transport genes were normalized to 16-s values and assessed by comparing the relative quantity of the reference mRNA in the absence of antibiotics. The qRT-PCR experiments were performed in duplicate and three biological replicates. A fold-change value of one indicates that the gene expression level is similar to the nonantibiotic-exposed control, while a fold-change less than one represents downregulation and greater than one is considered to be upregulated.

### 2.8. Biofilm Formation

The biofilm mass of sixteen MAB gene knockout clones and reference MAB 19977 was measured with the crystal violet staining method. Bacteria grown in the mid-log phase were resuspended at a concentration of 1 × 10^8^ CFU/mL in the Synthetic Cystic Fibrosis Sputum Medium (SCFM) established and prepared as previously described [28]. The uniformity for all the tested groups was confirmed with OD readings at OD_600_. A volume of 100 μL was placed in columns of a 96-well round-bottomed plate (Corning, Corning, NY, USA) for both the experimental and control groups (eight replicates) and incubated for one week at 37 °C (no agitation). The biofilm formation was quantified at 7 days by removing the supernatant from all wells and adding 125 μL of 0.1% crystal violet (CV) solution for 10 min at room temperature. After, the plates were rinsed three times with the distilled water and left to dry for 10 min. CV-stained biofilms were then solubilized with 125 μL of 30% acetic acid for 10 min. The samples were moved to new clear 96-well flat-bottomed plates, and the optical density was measured at OD_570_ in a plate reader (Epoch Microplate Spectrophotometer, BioTek, Winooski, VT, USA). The experiment was performed in three biological replicates.

### 2.9. MAB Survival Assay in THP-1 Human Macrophages

THP-1 cells were purchased from the ATCC (ATCC, TIB-202) and maintained in RPMI-1640 medium (Corning, Corning, NY, USA) supplemented with heat-inactivated 10% fetal bovine serum (FBS; Gemini Bio, Sacramento, CA, USA), l-glutamine, and 25-mM HEPES (Corning, Corning, NY, USA) at 37 °C and in an atmosphere of 5% CO_2_. For differentiation, the THP-1 cells were treated with 20 ng/mL of Phorbol 12-myristate 13-acetate (PMA; Sigma-Aldrich, St. Louis, MO, USA). Approximately, 3 × 10^5^ cells/well were seeded in 96-well plates and replenished 24 h later with new media. The THP-1 cells rested for an additional 48 h to allow for the formation of the >80% confluent monolayer. Macrophages were infected with MAB mutants and the wild-type MAB 19977 with a multiplicity of infection (MOI) of 1 bacterium to 1 cell and incubated at 37 °C in an atmosphere of 5% CO_2_. After 2 h of infection, the cells were washed three times with HBSS and treated with 200-μg/mL AMK for 2 h to remove the extracellular bacteria. The monolayers were lysed at 2 h (Baseline), 24 h, 72 h, and 120 h post-infection with 0.1% Triton X100, serially diluted with HBSS, and plated on 7H10 agar plates for viable bacterial CFU counts. We checked the integrity of the monolayers every day, as well as plated supernatants from each time point to determine the presence of extracellular bacteria, if any. The reference MAB 19977 was used as a control, and the infection percentage and survival rates were compared between the mutant strains and control at each time point.

### 2.10. The Complementation of MAB Mutants with Function Genes

To confirm the direct link of the gene inactivation to the observed phenotypes, the efflux pump-related mutants (*MAB_0937c* and *MAB_1137c*) were complemented with the functional genes as previously published [29]. Briefly, using the gene-specific primers, *MAB_0937c* and *MAB_1137c* genes were PCR-amplified from the DNA of MAB 19977, cloned into the pMV306 chromosomal integration plasmid, and transformed into the mutant clones. The transformant clones were plated and selected in the presence of 50-μg/mL apramycin, and the presence of functional genes was confirmed with PCR. The MAB wild-type strain with a naked plasmid served as the control.

### 2.11. Antibiotic Killing Kinetics in Human Macrophages

THP-1 monocytes were differentiated in the 96-well plates as descried above, and the macrophage monolayers were infected at an MOI of 1:1 with one of the following strains: the wild-type MAB 19977, knockout mutant *MAB_0937c*^Δ^, complemented mutant *MAB_0937c*^(+)^, knockout mutant *MAB_1137c*^Δ^, or complemented mutant *MAB_1137c*^(+)^. The infected monolayers were treated with AMK 32 μg/mL or no antibiotic and replenished with a new media at day 3. The THP-1 monolayers were lysed at 1 h to record the invasion rates. To determine the number of viable intracellular bacteria over time, the cell lysates were obtained at days 1, 3, and 5; serially diluted; and plated on 7H10 agar plates for the CFU counts.

### 2.12. Statistics

Statistical comparisons between the control and experimental groups were performed using a Student’s *t*-test. The graphical outputs were expressed as the mean ± standard deviation and were created with GraphPad Prism 9.0 software. The results from three biological replicates were used unless otherwise indicated. The * *p*-values of <0.05 and ** *p* < 0.01 were considered statistically significant and are indicated in the figure legends.

## 3. Results

### 3.1. MAB Gene Knockout Mutants Associated with Increased or Decreased Susceptibility to Antibiotics

In attempts to identify MAB genes that play a role in an antibiotic intrinsic resistance, the transposon library of gene knockout mutants was evaluated in comparison to the parental MAB 19977 for increased or decreased susceptibility to the aminoglycoside, macrolide, and cephalosporin groups of beta-lactam antibiotics AMK, CLA, and FOX, respectively. We hypothesized that the loss of function in the genes related to the inactivation or removal of antimicrobials from bacterial cells will display an increased sensitivity to antibiotics, and it would result in a decrease of MIC concentrations when compared to the wild-type strain. On the other hand, if genes encoding proteins such as porins, enzymes, or possibly elements regulating the uptake of substrates within the cells that are potentially involved in the drug influx mechanism, these subsets of mutants will display an increased resistance to antimicrobials and will display higher BC concentrations than the wild-type strain. We established 134 mutants, of which 96 displayed decreased MIC and 41 increased BC concentrations, sequenced using ligation-mediated PCR and analyzed to determine the potential functions of the knockout genes (Appendix A). In addition, we categorized the sequenced genes based on the functional classes (Figure 1). The categories included transcription regulators, enzymes associated with metabolic reactions, proteins involved in the influx and efflux transport mechanisms, and many hypothetical proteins with unknown functions. Table 3 details the list of mutants with gene functions relating to cell surface integral membrane proteins and transport mechanisms that we further characterized in this study.

### 3.2. MAB Surface Factors Associated with Differential Antibiotic Susceptibility Patterns

Sixteen gene knockout mutants related to the surface transport systems that exhibited varied antibiotic susceptibility versus the parental MAB 19977 were selected and further assessed for the MIC and BC concentrations of AMK, CLA, and FOX using the broth microdilution method.
Table 4
details the
differential antibiotic susceptibility patterns in MAB mutants highlighting distinguished phenotypic changes depending on the mutants and antibiotics tested when compared to the wild-type strain.

### 3.3. MAB Surface Factors Associated with the Biofilm Formation

Biofilms are one of the major contributing factors of mycobacterial resistance to antibiotics [30]. The biofilm formation process relies on the ability of mycobacteria to transport the critical components of the matrix, including extracellular DNA (eDNA), proteins, mycolic acids, glycopeptidolipids, and other lipids, out of the cell [31]. Biofilms also confer an improved attachment ability of the bacteria, and the presence of extracellular polysaccharides increases the difficulty of the desiccation process within the biofilm matrix [32]. The identification of the transport genes involved in the biofilm formation will play an additional role in improving the antibacterial therapy for MAB infections.

To determine if some of the transport-related genes identified during our screen also contributed to the biofilm formation process, the knockout mutants of the export mechanisms were tested for biofilm matrix integrity and decreased efficiency for biofilm formation in comparison to the wild-type control. The biofilm formation of the sixteen selected mutants and parental MAB 19977 were assessed in the synthetic cystic fibrosis sputum media (SCFM) and visualized using the crystal violet staining method at the one-week time point. Figure 2 shows the percentage of biofilm formation over the wild-type control in SCFM, where the biofilm of the reference MAB 19977 in HBSS served as a negative control. Overall, the *MAB_1171c*, *MAB_2787c*, and *MAB_4036* gene knockout mutants showed an approximately 50% decrease in the biomass when compared with the positive control at the one-week time point. We did not observe significant changes in the biofilm mass formation for the rest of mutants (Figure 2).

### 3.4. The Ability of MAB Transport System Mutants to Infect and Grow Intracellularly in THP-1 Human Macrophages

To assess the effect of gene deficiency on the bacterial ability to infect, survive, and grow in the intracellular environment of macrophages, sixteen selected MAB mutants of transport systems were utilized in the THP-1 macrophage infection assay, and the bacterial growth rates were calculated in comparison to the wild-type infection over 5 days. A 1-h time point invasion (baseline) demonstrated similar uptake rates for all mutants within the host cells (Figure 3A). While some of the mutants exhibited decreased survival in THP-1 cells, none of the mutants displayed statistical significance at the intracellular survival when compared to the wild-type infection of phagocytic cells (Figure 3B).

### 3.5. The Expression of Surface Associated Transport Genes across MAB Clinical Isolates of Cystic Fibrosis Patients

To establish the relevance of the selected transport genes in the mechanism of MAB intrinsic resistance, we evaluated the expression levels of the selected genes across nine MAB clinical isolates during antibiotic exposure, while no antibiotic treatment served as a control. In the initial screening, each clinical strain was cultured in 7H9 broth containing MIC concentrations of AMK and CLA, and the gene expressions were evaluated after 24-h exposure (Appendix A). Since it takes over 5 days to kill MAB with bactericidal concentrations of drugs, we hypothesized that the efflux pumps and transport mechanisms associated with MAB tolerance to antibiotics will likely be expressed constitutively at least in the initial 24 h of drug exposure. The fold-changes were calculated for each condition and strain in comparison to the control. The data were normalized with the conserved 16S rRNA gene Ct values.

Overall, the analysis identified large variabilities for each gene expression level across nine clinical isolates, which possibly may relate to different MIC patterns of these strains (shown in Appendix A and can also explain no change in the gene expression levels where resistance or increased MICs are observed. The *MAB_4237c*, *MAB_1839*, *MAB_0734*, *MAB_1137c*, and *MAB_4117c* genes showed a consistent increase in gene expression during exposure to both antibiotics for most clinical isolates (Appendix A). The expression of the selected four genes, three related to the efflux pumps and one ABC transporter, was also evaluated at an early (8 h) time point during AMK exposure. While *MAB_0937c, MAB_1137c*, and *MAB_4237* demonstrated a consistent increase in gene expression at the 8-h time point for all clinical isolate strains, including the reference strain (Figure 4), the *MAB_4117c* gene showed an increased expression for most of the clinical isolates except NR 44273 strain 4529, DNA00703, and DNA01715 (Figure 4).

### 3.6. The Functional Disruption of Efflux Pumps Increases the Antibiotic Efficacy against MAB of the Intracellular Phenotype

Previous studies have shown that inhibiting the function of drug exporter *mmpL* genes reduced the intrinsic resistance to antibiotics [33]. We evaluated two MmpL efflux pump-deficient mutants (*MAB_0937c*^Δ^ and *MAB_1137c*^Δ^) associated with increased sensitivity to antibiotics and tested if the functional disruption of these genes would impact the pathogen’s survival ability during the AMK (32 μg/mL) treatment of infected human macrophages, suggesting an improved efficacy of the used antibiotics. The parental strain and complemented clones of *MAB_0937c*^(+)^ and *MAB_1137c*^(+)^ were also evaluated. At first, we assessed the intracellular growth of gene knockout mutants and complement clones in THP-1 cells without antibiotic treatment to ensure that the generated clones did not have an attenuation phenotype. As shown in Figure 5A, all the tested clones displayed similar growth phenotypes over 5 days of infection as the wild-type strain. However, the CFU counts of both *MAB_0937c*^Δ^ and *MAB_1137c*^Δ^ mutants treated with AMK demonstrated a significant reduction in viable bacteria at 72 h and 120 h post-infection when compared with the complemented clones and the wild-type strain (Figure 5B). It was also observed that the complementation of both mutants restored the intracellular survival rates of the gene knockout mutants similarly to the MAB 19977 control, suggesting the direct link of efflux pump genes *MAB_0937c* and *MAB_1137c* to antibiotic resistance.

## 4. Discussion

*Mycobacterium abscessus* subsp. *abscessus* has been long-recognized as one of the most antibiotic-resistant NTMs; however, the basis of intrinsic resistance is still poorly understood, making its treatment extremely challenging in clinics [5,7]. The development of effective compounds that can circumvent the natural resistance mechanisms of MAB is a daunting task and requires the identification and characterization of bacterial genetic factors that encode enzymes with drug or drug target-modifying properties [5,34,35], multidrug export systems that have been proven to be a reason for the efficient efflux of the aminoglycoside, macrolide, and quinolone classes of antibiotics from cells [9] and the key metabolic enzymes promoting a metabolic shift in persistence phenotypes where the current antibiotics are ineffective [25]. This study aimed to identify the MAB determinants of intrinsic resistance associated with a tolerance to the aminoglycoside, macrolide, and cephalosporin classes of antibiotics AMK, CLA, and FOX, respectively, while validating some targets of the MAB efflux system that work in synergy with the existing antibiotics and demonstrate improved clinical efficacy.

The mechanisms of mycobacterial intrinsic resistance are multifold and largely associated with a highly impermeable cell wall and the presence of the drug efflux pumps, significantly reducing the concentration and, therefore, efficacy of antimicrobials. The primary physiological role of efflux pumps is maintaining the balance of nutrient metabolites and environmental toxins for bacteria, but it also allows the pathogen to extrude a broad range of antibiotic drugs [36,37,38]. The *MAB_0937c*, *MAB_1137c*, and *MAB_4117c* gene-deficient mutants identified during our MIC screen had a loss of function associated with the MmpL family efflux systems. The MmpL family proteins belong to multidrug resistance pumps termed Resistance-Nodulation-Cell Division (RND) permeases involved in the export of lipid components across the cell envelope [33,39,40]. MmpL membrane proteins are hydrophobic, spanning the membrane with two extra cytoplasmic domains. The presence of antibiotics can stimulate an increased expression of MmpL transporters [41,42]. These transporters utilize the electrochemical gradient of protons across the cell membrane to expel drugs and are highly regulated [43]. Previous evidence shows that the deletion of specific *mmpL* genes leads to a decrease of pathogenicity and sustained bacterial growth of *M. tuberculosis* in mice [40]. Our results show that *MAB_0937c*, *MAB_1137c*, and *MAB_4117c* export genes are highly expressed in most clinical isolates of MAB when exposed to antibiotics. While all three mutants contributed to antibiotic susceptibility, the *MAB_0937c* and *MAB_1137c mmpL* gene-deficient clones were associated with a decreased tolerance to antibiotics, and *MAB_4117c* (*mmpS*) exhibited an increased resistance to the BC concentration. Furthermore, *MAB_0937c* is highly homologous to the *mmpL*10 gene of *M. tuberculosis* known to export cell wall components such as di/poly-acyl trehalose (DAT/PAT) and is important for bacterial pathogenicity [38].

Of the mutants sequenced and analyzed, 26 had genes found to be associated with the cell membrane and transport systems. Seven mutants (*MAB_1839*, *MAB_3384c*, *MAB_2435, MAB_2785, MAB_2787c, MAB_4036*, and *MAB_4237c*) identified in our study belong to the ABC transporters. The ABC transporters are a large family of proteins involved in the transport of various compounds, such as sugars, ions, peptides, and more complex organic molecules [44], and have been shown to use ATP hydrolysis in the transport of antibiotics outside the cell [45]. Our research indicates that the loss of function in ABC transporter genes results in an increase of the susceptibility of MAB to FOX, AMK, and CLA and suggests the relevance of these transporter genes to the intrinsic resistance of the pathogen. In addition, two putative lipoproteins encoded by the *LpqW* (*MAB_1315*) gene and predicted membrane *MAB_1418* located downstream of the *LprC* (*MAB_1417*) lipoprotein (functioning as an operon) and identified in our study were also linked with MAB tolerance mechanisms. Lipoproteins are integral components of the mycobacterial cell wall. Previous studies showed that the lipoprotein LprB displays an increased production in *M. avium* subsp. *hominissuis* biofilms when exposed to antibiotic treatment [46]. LprB has also been shown to play a role in *M. avium* subsp. *hominissuis* virulence and survival in vitro [46].

In addition, our study identified the BC mutant of the *MAB_0734* (MspA) porin gene displaying resistance to bactericidal concentrations of all three antibiotics tested. The cell surface influx system, such as porin channels, mediates the diffusion of small hydrophilic molecules. Previous studies performed in *M. smegmatis* and *M. tuberculosis* demonstrated that the addition of the *mspA* gene to *M. tuberculosis* decreased the MIC to a wide range of antibiotics and significantly increased the bacterial susceptibility [47]. These findings show direct evidence of the role of porins that influence the microbial susceptibility to antimicrobials.

The second-largest functional category identified in this study are metabolic enzymes. For example, *MAB_0304* and *MAB_1393c* genes belong to oxoglutarate decarboxylase enzymes that cleave the carbon–carbon bond in alpha-ketoglutarate in the citric acid cycle [48]. The alpha-ketoglutarate dehydrogenase (KDH) complex is a central regulatory point of aerobic energy metabolism. Studies on *M. tuberculosis* show that this enzyme is regulated by the acyltransferase-like domain and the concentration of acetyl-CoA [49]. Therefore, targeting this class of enzymes could potentially impact the intracellular survival and virulence of MAB in the host due to the essential role of this protein in cellular respiration.

The gene *MAB_1865* encodes for a fatty acid–CoA ligase of the FAA1 COG1022 family involved in the fatty acid biosynthesis and metabolism. Long-chain acyl-CoA synthetase (AMP-forming) catalyzes an ATP-dependent two-step reaction to activate a carboxylate substrate as an adenylate and then transfers the carboxylate to the pantetheine group of either coenzyme A or an acyl carrier protein. A study by Daniel at al. showed that an acyl-CoA synthetase (Rv1206) resembled a mammalian fatty acid transport protein, and the expression and bacterial cell enrichment with FACL6 (Rv1206) were associated with a *M. tuberculosis* dormancy state in vitro and, also, capable of modulating triacylglycerol accumulation and stimulating fatty acid uptake in *E. coli* [50]. This study highlights an importance of long-chain acyl-CoA synthetase enzymes, playing a crucial role in *M. tuberculosis* dormancy and, subsequently, resulting in developing a resistance mechanism to antibiotics.

Interestingly, our screen discovered the MIC mutant *MAB_2979* of the peptide–methionine (R)–S-oxide reductase (*selR*) gene. The *selR* domain proteins are known to represent methionine sulfoxide reductase enzymes that reduce methionine (R)-sulfoxide back to methionine, which is a necessary reaction to regulate the biological processes to cope with oxidative stress [51]. This information may explain some of new mechanisms on how MAB can overcome the oxidative stress encountered within the intracellular environment of phagocytic cells while also contributing to the intrinsic resistance fitness of the pathogen.

Our screen identified a handful of regulatory elements associated with MAB susceptibility to antibiotics. Genes that encode the GntR, LysR, and TetR/AcrR transcription factors belong to multidrug efflux regulators and have been shown to be directly involved in regulating the function of efflux systems [52,53,54,55,56]. A study involving an *M. tuberculosis Rv1152* gene of the GntR family regulators demonstrated that the overexpression of this gene in *M. smegmatis* decreased the susceptibility of mycobacteria to vancomycin [57]. It was also found that Rv1152 negatively regulated several vancomycin responsive genes, highlighting the importance of GntR transcription factor function for vancomycin susceptibility [57]. Furthermore, the gene *MAB_0161* belongs to the LysR domain and is downstream from an ABC transporter gene. *MAB_0068* is also a transcription regulator located upstream from *MAB_0069*, a significant facilitator of transporter family-related proteins. Gene *MAB_0068* shows functionality for a putative transcription regulator in the GntR family. The location of this transcription regulator adjacent to a significant transporter creates a possibility that *MAB_0068* has ties for regulating the transport mechanisms.

We also discovered the transcription regulator genes *MAB_4710c, MAB_1589, MAB_1881c*, and *MAB_2061c* that belong to the TetR family. The TetR family regulators act as chemical sensors to monitor the cellular environment [52,58]. It has been found that MAB TetR regulator gene *MAB_2299c* controls the expression of two distinct MmpS–MmpL efflux pumps involved in the cross-resistance to clofazimine and bedaquiline in MAB [19]. In addition, the multidrug-binding proteins AcrR of *Escherichia coli* and CmeR from *Campylobacter jejuni* are members of the TetR family of transcription repressors that regulate the expression of AcrAB and CmeABC efflux pumps, respectively [58]. The multidrug-binding repressor QacR of *Staphylococcus aureus* recognizes a combination of drugs by using multiple proximal and distinct drug-binding sites, and this recognition helps with the upregulation of the genes necessary for antibiotic resistance [59]. Moreover, the study by Wagner at al. demonstrated that the overexpression of the TetR family regulator Ms4022 activates the expression of seven transport-related genes and increases the drug resistance of bacteria to rifampicin. Contrary, the *MSMEG_4022* gene knockout mutant exhibited a decreased resistance to rifampicin [55]. These findings highlight an important function that the TetR family genes may have in the transcriptional regulation of MAB intrinsic resistance to antibiotics.

To validate the significance of the identified MAB genes in the resistance to antibiotics, we focused on sixteen surface-related transport genes and investigated their gene expression levels across nine clinical isolates during exposure to three functional groups of antibiotics. Appendix A details a differential expression of the transport genes obtained using RT-qPCR. The activation of MAB surface export genes during antibiotic exposure was obvious at early time points of antibiotic treatment for mmpL genes *MAB_0937c* and *MAB_1137c*, the *MAB_4117c* gene of the mmpS efflux pump, and ABC transporter *MAB_4237c* when compared to the nonantibiotic treatment control.

Moreover, since bacterial transport systems are also involved in the export of different types of constituents for the formation of a mycobacterial biofilm matrix [60,61], and these structures create an additional complexity for drug penetration, further decreasing the antibiotic efficacy [25], we tested the MAB transport-related mutants for deficiency in forming the biofilm matrix. Our results demonstrated that ABC transporter genes *MAB_2787c, MAB_1171c*, and *MAB_4036* are likely associated with the transport of biofilm constituents out of the cell, since gene knockout mutants have significant decreases in biofilm mass.

The host environment is another important factor that stimulates metabolic changes and facilitates cell surface remodeling in intracellular bacteria that reside within phagosome vacuoles [62]. These phenotypic changes subsequently promote mycobacterial tolerance phenotypes that influence the antibiotic treatment efficacy [25,46,63]. To test if the transport-related mutants may also display an attenuation phenotype for intracellular growth, we monitored MAB survival within the THP-1 human macrophages over five days of infection. None of the mutants were found to be deficient in intracellular growth. However, when we tested two selected MmpL efflux pump mutants of the *MAB_0937c* and *MAB_1137c*-deficient genes for survival assays in combination with AMK treatment, we could demonstrate a direct relationship of the transport genes in increasing the antibiotic efficacy and synergy, accelerating the MAB clearance in infected macrophages. Conversely, when the genes were functionally complemented, this increased the tolerance phenotype of the pathogen to the selected antibiotic and, subsequently, exhibited survival in a similar manner as the parental strain.

Our research identified MAB genetic determinants of export systems, some of them already defined as multidrug efflux pumps and ABC transporters, metabolic enzymes shown to promote bacterial metabolic remodeling in persistent phenotypes, and regulatory elements, all associated with a susceptibility to clinically relevant drugs and some involved in biofilm formation as well. We also demonstrated that, if the efflux elements of MAB are inactivated, this significantly influences the bacterial susceptibility to antibiotics and synergizes the killing of intracellular bacteria in the host cells. Future studies are encouraged to understand the functions of many of the uncharacterized genes identified in this study and their relevance to MAB intrinsic resistance.

## Figures and Tables

**Figure 1 microorganisms-09-02527-f001:**
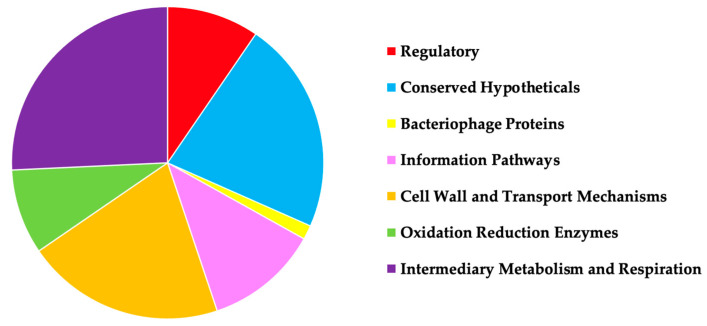
A pie chart displays MAB 19977 transposon mutants identified in this study that are grouped based on the protein functional classes. The gene knockout mutants were sequenced with LM-PCR, and transposon insertions were analyzed by blasting obtained nucleotide sequences (adjacent to the transposon) against the reference genome of MAB 19977. The gene domains/motifs of the putative proteins were analyzed using the CD-Search tool available at NCBI and by aligning proteins to the well-characterized genome of *M. tuberculosis* H37Rv.

**Figure 2 microorganisms-09-02527-f002:**
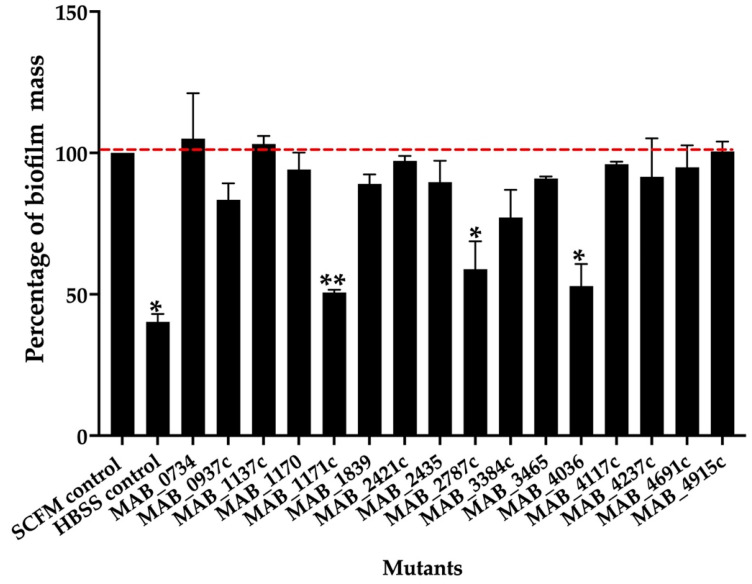
The percentage of biofilm mass for MAB mutants belonging to the surface transport systems. The biofilm mass of the MAB wild-type and transposon mutants were quantified at day 7 with 0.1% crystal violet staining and solubilization of 30% acetic acid, and the measurements were recorded at OD_570_. The percentage of biofilm formation was normalized to the positive control, representing the MAB 19977 biofilm mass in SCFM (dashed red line), while the MAB 19977 biofilm mass in HBSS represents the negative control. Data represents the mean ± SD of wells assayed in duplicate and with three biological replicates. Significant differences were observed between the positive biofilm of MAB 19977 in SCFM and the negative control in HBSS (* *p* < 0.05), *MAB_1171c* (** *p* < 0.01), *MAB_2787c* (* *p* < 0.05), and *MAB_4036* (* *p* < 0.05).

**Figure 3 microorganisms-09-02527-f003:**
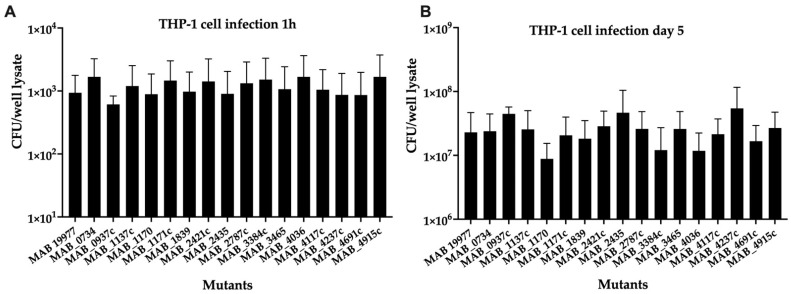
The intracellular survival of MAB mutants associated with surface transport systems. The THP-1 monocytes were differentiated with 20-ng/mL PMA in 96-well plates and rested for 3 days to form confluent monolayers. The macrophages were then infected with one of 15 mutants or the wild-type MAB 19977 (control) at a MOI of 1:1. While an infection rate was measured at 1 h (**A**), the survival rates were determined at 5 days post-infection (**B**) by lysing cells with 0.1% Triton X100 and plating lysates on 7H10 agar plates for viable bacterial CFU/well counts.

**Figure 4 microorganisms-09-02527-f004:**
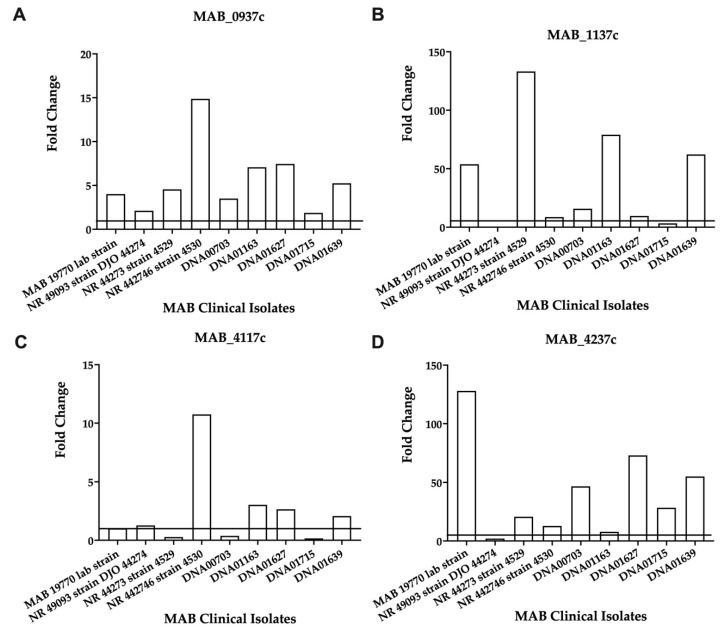
The fold changes in the efflux pump and ABC transporter gene expressions in the clinical isolates of MAB exposed to AMK. Each clinical strain at 3 × 10^8^ CFU/mL was matched to the McFarland Standard 1.0 and inoculated with 1× MIC of AMK. The RNAs were extracted after 8 h, and the DNase was treated and used for cDNA synthesis, as described in the Materials and Methods. The gene expressions were evaluated with qRT-PCR. (**A**) *MAB_0937c*, MmpL drug efflux pump; (**B**) *MAB_1137c*, MmpL efflux pump; (**C**) *MAB_4117c*, efflux pump related MmpS; (**D**) *MAB_4237c* encodes for an ABC transporter.

**Figure 5 microorganisms-09-02527-f005:**
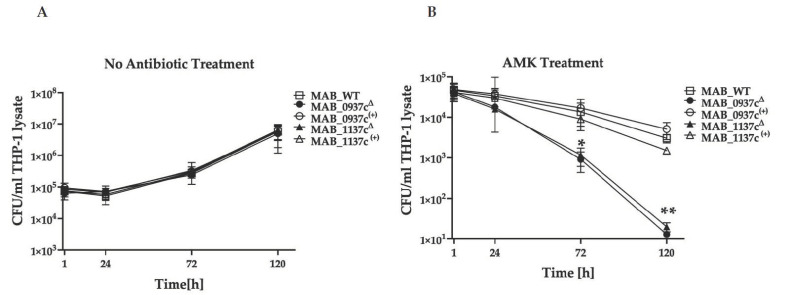
The improved antibiotic efficacy against infections of MAB efflux pump-deficient mutants in THP-1 macrophages. (**A**) The intracellular growth dynamics of *MAB_0937c*^Δ^ and *MAB_1137c*^Δ^ gene knockout mutants and *MAB_0937c*^(+)^ and *MAB_1137c*^(+)^ complemented clones without AMK treatment in comparison to the wild-type MAB 19977 were recorded with CFU/mL counts over 5 days of infection and by lysing THP-1 cells at the 1-h, 24-h, 72-h, and 120-h time points. (**B**) Intracellular survival rates of gene knockout and complemented clones during AMK treatment measured with viable bacterial CFU counts at 1 h, 24 h, 72 h, and 120 h post-infection. Data represent the mean ± SD of two independent experiments, each completed in triplicate. * *p* < 0.05 and ** *p* < 0.01 between the complemented clone and mutant at the corresponding time points.

**Table 1 microorganisms-09-02527-t001:** Antibiotic concentrations used in the MAB mutant library susceptibility testing.

Antibiotic	MAB 19977	
	MIC (μg/mL)	BC (μg/mL)
AMK	<16	32
CLA	<2	8
FOX	<16	128

AMK: amikacin, CLA: clarithromycin, and FOX: cefoxitin.

**Table 2 microorganisms-09-02527-t002:** Primers used in this study for the qRT-PCR analysis of the transport genes in the MAB clinical isolates.

MAB Gene	Sequence (5′-3′) Forward	Sequence (5′-3′) Reverse
*MAB_0734*	CTCGCCAACGCGAGTAATTC	CAGCCCAGCTGATAACCCAT
*MAB_0937c*	ATCAGTGATGTCGCCAAGGG	AGCGTCAACAGAACGACGAT
*MAB_1137c*	GTGATATCGCTGCTCGGTGA	CGCGACGATGAAAATCGCAT
*MAB_1170*	CTGTGATTGGGGCGCTATTG	TTCTTCGTCTTAGCCGCCG
*MAB_1171c*	GCCAACGCATCCAAACCTAC	GATTCTGCGGTAGAGCCGAT
*MAB_1839*	TGGGGAATTCGTTACGCGAG	CCACGCCAACTTCACAAAGG
*MAB_2421c*	CTCGACTACGAGGGGAAAGC	GGTCGTCCATTTGTCACCGA
*MAB_2435*	GAGATTCGTCTCGGCGATCA	GTCAGATGGGGAAACAGCGA
*MAB_2787c*	GGGCTCACCTACTCCTCGTA	TTGGTTCTCCGACTTGGTGG
*MAB_3384c*	AGGGTTCAGTCACGCATTGT	GTAACGCCCGTCGATCATCT
*MAB_3465*	GGAGCTACGCACACCAAAGA	GCGATTCAGCACTTCCCTGA
*MAB_4036*	GACCACCCGCAATGTACAGA	TGCATGATGGTTCCGGTTGA
*MAB_4117c*	CTCCACGGCGTATTCGGTAA	ACAGCGGATCACCATTGACA
*MAB_4237c*	AATCTGACTGTGCACCAGGG	ATCAGCTCACCGTCGATACG
*MAB_4691c*	TGCGATAGACGACGTACTGC	CAATGGACCGGTGATGGTGA
*MAB_4915c*	ATGTCGAAATGCGATGGGGT	GCCGATAACACTACGGGTCC
16S	GGCTAACCATCCGTCTCTGG	CGGAAGAAAGTCGTCGGTCA

**Table 3 microorganisms-09-02527-t003:** The selected MAB 19977 gene knockout mutants of cell wall transport systems associated with an increased (BC) or decreased (MIC) susceptibility to antibiotics.

Gene	Function	Conserved Domain/Notes	MIC/BC
*MAB_0734*	MspA membrane porin	Contains a signal peptide	BC
*MAB_0937c*	MmpL membrane drug exporter protein	Transport protein with a role in drug resistance	MIC
*MAB_1137c*	Putative MmpL membrane protein	Possible role in drug resistance	MIC
*MAB_1170*	Putative membrane transporter protein.	TauE, sulfite exporter. Integral membrane protein involved in the transport of anions across the cytoplasmic membrane during taurine metabolism as an exporter of sulfoacetate.	MIC
*MAB_1171c*	Conserved hypothetical protein	Integral membrane transporter protein	MIC
*MAB_1839*	Diguanylate cyclase/phosphodiesterase	GGDEF Diguanylate-cyclase/ABC2_membrane superfamily regulates cell surface adhesion in bacteria/ABC-2-type transporter	MIC
*MAB_2421c*	Hypothetical protein	PknH_C-like extracellular domain	BC
*MAB_2435*	Molybdenum ABC transporter ModC, ATP-binding protein	P-loop_NTPase superfamily	MIC
*MAB_2787c*	Probable molybdenum ABC transporter, periplasmic	Periplasmic-binding protein type 2 superfamily	MIC
*MAB_3384c*	Putative ABC transporter, ATP-binding protein	ModF ABC-type molybdenum transport system	MIC
*MAB_3465*	Putative sulfate transporter/anti-Sigma factor		MIC
*MAB_4036*	Conserved hypothetical protein	ABC_6TM_exporters superfamily: Six-transmembrane helical domain of an uncharacterized ABC exporter	MIC
*MAB_4117c*	Putative membrane protein	MmpS family protein with a potential role in the transport of MmpL substrates	BC
*MAB_4237c*	Putative amino acid ABC transporter, ATP-binding protein	GlnQ	MIC
*MAB_4691c*	Probable non-ribosomal peptide synthetase PstA	Sulfate Transporter and anti-Sigma factor antagonist	MIC
*MAB_4915c*	Hypothetical protein	PknH_C-like extracellular domain	BC

MIC: minimal inhibitory concentration; BC: bactericidal concentration.

**Table 4 microorganisms-09-02527-t004:** The antibiotic susceptibility of MAB gene knockout mutants associated with the surface transport systems and MAB clinical isolates.

Bacterial ID		MIC for Broth Dilution in 7H9 (μg/mL)		
		AMK	CLA	FOX
**MIC mutants**	*MAB_0937c*	6	0.5	4
	*MAB_1137c*	6	0.5	4
	*MAB_1170*	12	0.5	16
	*MAB_1171c*	16	0.5	4
	*MAB_1839*	12	0.5	8
	*MAB_2435*	10	0.5	16
	*MAB_2787c*	10	2	4
	*MAB_3384c*	10	0.5	16
	*MAB_3465*	10	2	16
	*MAB_4036*	10	0.5	4
	*MAB_4237c*	10	2	8
	*MAB_4691*	10	0.5	4
**BC mutants**	*MAB_0734*	128	8	32
	*MAB_2421c*	128	4	32
	*MAB_4117c*	64	4	32
	*MAB_4915c*	128	8	32
**Clinical isolates**	MAB 19977	16	2	16
	NR 49093 strain DJO 44274	>256	>32	16
	NR 44273 strain 4529	>128	>32	16
	NR 44273 strain 4529	16	4	16
	DNA00703	16	4	32
	DNA01163	16	4	32
	DNA01627	16	16	32
	DNA01715	16	4	32
	DNA01639	16	2	32

## Data Availability

Not applicable.

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
