# Peer review of "Mycobacterium abscessus Genetic Determinants Associated with the Intrinsic Resistance to Antibiotics"

_microorganisms, 2021, doi:10.3390/microorganisms9122527_

Round 1

Reviewer 1 Report

Major comments

Lines 14–45 Abstract is too long; looks like Conclusion chapter. I recommend according to the journal recommendations to reduce abstract for only 200 words.

Lines 46–47 Keywords: M. abscessus subsp. abscessus; intrinsic resistance; virulence factors; efflux pumps; bio-46 film; macrophage „add the most common used term: rapidly groing mycobacteria“;

Introduction: Please, add one paragraph describing the whole comples (i.e. you can quote reference: Minias A, Żukowska L, Lach J, Jagielski T, Strapagiel D, Kim SY, Koh WJ, Adam H, Bittner R, Truden S, Žolnir-Dovč M, Dziadek J. Subspecies-specific sequence detection for differentiation of Mycobacterium abscessus complex. Sci Rep. 2020 Oct 2;10(1):16415 or others).

Minor comments

Mycobacterium …. In italic through the whole list of References (in a total of 43 references!).

Through the whole text use abbreviated “min” instead of “minutes”.

Through the whole text use abbreviated “h” instead of “hours”.

Line 14 start with the full name: Mycobacterium, not with abbreviated M.

Line 14 “mycobacterium” change for Mycobacterium

Line 15 “individuals” change for patients

Line 20 “resistance, we screened…“ change for „ resistence; we screened…“

Line 54 “2.6% - 13.0 %“ change for 2.6–13.0%

Line 59 add “therapeutic” regimens

Line 115 “The reference strain M. abscessus subsp. abscessus strain 19977“ change for „The reference Type strain M. abscessus subsp. abscessus (ATCC 19977)“

Line 117 “clinical strains“ change for „clinical isolates“

Line 134 “MAB strain 199770“ change for “MAB Type strain 199770“

Line 158 “wild-type MAB 19977” change for “wild-type strain MAB19977“

Line 166 under the Table 1 add the explanations of abbreviations

Line 169 “dH2O“ change for “dH2O”

Line 177 “100-150 ng“ change for „100–150 ng“

Line 192 add producer address

Line 209 “wild-type control” change for “wild-type strain MAB 19977 used as control“; and please, change it to the same form through the whole text.

Line 229 (Invitrogen): add the producer including address

Line 266 “MAB 19977 strain“ change for „MAB 19977 Type strain“

Line 340 change the Figure 1 to normal chart (without spatial effect)

Line 348 under the Table 3 add the explanations of abbreviations

Line 355 “table 4” change for “Table 4”

Line 358 under the Table 4 add the explanations of abbreviations

Line 372 create new paragraph (the previous text describes the biofilm formation process, not your results)

Line 375 “figure 2” change for “Figure 2”

Line 408 remove one dot

Line 416 “Supplemental table 2“ change for „Supplemental Table 2” thgrough the whole text (see also lines 423, 426, etc.)

Line 421 create new paragraph

Line 469 “M. abscessus subsp. abscessus“ change for „Mycobacterium abscessus subsp. abscessus“

Line 491 create new paragraph

Line 517 remove [45] (it is quoted twice in one paragraph)

Author Response

We would like to thank the reviewer for thorough review and suggestions that majority of them have been incorporated in the paper. All changes are mark in red.

Lines 14–45 Abstract is too long; looks like Conclusion chapter. I recommend according to the journal recommendations to reduce abstract for only 200 words.

A: We significantly reduced the abstract size.

Lines 46–47 Keywords: M. abscessus subsp. abscessus; intrinsic resistance; virulence factors; efflux pumps; bio-46 film; macrophage „add the most common used term: rapidly groing mycobacteria“;

A: We added the key word “rapidly growing mycobacteria” but kept M. abscessus (as a short name) because there are many fast-growing mycobacterial species that are pathogens but significantly differ from M. abscessus in virulence, genetics and ect. We also added “antibiotic treatment” as additional key word.

Introduction: Please, add one paragraph describing the whole comples (i.e. you can quote reference: Minias A, Żukowska L, Lach J, Jagielski T, Strapagiel D, Kim SY, Koh WJ, Adam H, Bittner R, Truden S, Žolnir-Dovč M, Dziadek J. Subspecies-specific sequence detection for differentiation of Mycobacterium abscessus complex. Sci Rep. 2020 Oct 2;10(1):16415 or others).

A: As suggested by a reviewer, we added the reference in the introduction (line 40). It is #2 in the reference list.

Minor comments

Mycobacterium …. In italic through the whole list of References (in a total of 43 references!).

A: We italicized Mycobacterium in all references.

Through the whole text use abbreviated “min” instead of “minutes”.

A: We replaced “minutes” to “min” throughout the text.

Through the whole text use abbreviated “h” instead of “hours”.

A: We replaced “hours” to “h” throughout the text.

Line 14 start with the full name: Mycobacterium, not with abbreviated M.

A: We made this change on the line 14.

Line 14 “mycobacterium” change for Mycobacterium

A: mycobacterium (second one on the line 14) is used in the contest of genus and it should not be italicized, and no capital letter is required either.

Line 15 “individuals” change for patients

A: “immunocompetent” means having a normal immune response or same as being healthy, thus, individuals is more appropriate to use in this sentence for both “immunocompromised and immunocompetent”.

Line 20 “resistance, we screened…“ change for „ resistence; we screened…“

A: We made changes to this sentence Lines 19-20.

Line 54 “2.6% - 13.0 %“ change for 2.6–13.0%

A: Line 41: We made this change.

Line 59 add “therapeutic” regimens

A: Line 46: We added “treatment” regimens.

Line 115 “The reference strain M. abscessus subsp. abscessus strain 19977“ change for „The reference Type strain M. abscessus subsp. abscessus (ATCC 19977)“

A: Line 102: We added “Type” in this sentence and defined this strain as “MAB 19977”.

Line 117 “clinical strains“ change for „clinical isolates“

A: Line 105. We made this change.

Line 134 “MAB strain 199770“ change for “MAB Type strain 199770“

A: Line 122: We defined MAB Type strain 199770 as MAB 19977 on the line 102 and used this name throughout the text.

Line 158 “wild-type MAB 19977” change for “wild-type strain MAB19977“

Line 146: We defined “the wild-type MAB 19977” on the line 102 and used it throughout the text.

Line 166 under the Table 1 add the explanations of abbreviations

A: Line 154: We removed full names of antibiotics from table and placed abbreviations below the Table 1.

Line 169 “dH2O“ change for “dH2O”

A: Line 157: changed 2 to a subscript.

Line 177 “100-150 ng“ change for „100–150 ng“

A: Line 165: we fixed the hyphen.

Line 192 add producer address

A: Line 181: Thermo Fisher Scientific, Waltham, MA USA added headquarters address

Line 209 “wild-type control” change for “wild-type strain MAB 19977 used as control“; and please, change it to the same form through the whole text.

A: Lne 198: We defined “the wild-type MAB 19977” on the line 102 and changed the text to “by comparing to the wild-type MAB 19977 as a control”.

Line 229 (Invitrogen): add the producer including address

A: Line 220: We added Invitrogen address Carlsbad California.

Line 266 “MAB 19977 strain“ change for „MAB 19977 Type strain“

A: We removed strain and used as defined MAB 19977.

Line 340 change the Figure 1 to normal chart (without spatial effect)

A: Line 329: We changed the Figure 1 as suggested.

Line 348 under the Table 3 add the explanations of abbreviations

A: Line 338: we added explanation for abbreviations under the table.

Line 355 “table 4” change for “Table 4”

A: Line 343: we added capitalization.

Line 358 under the Table 4 add the explanations of abbreviations

A: As abbreviations has been already defined under Tables 1 and 3, there is no need to repeat it again under Table 4.

Line 372 create new paragraph (the previous text describes the biofilm formation process, not your results)

A: Line 357: We added indent for new paragraph.

Line 375 “figure 2” change for “Figure 2”

A: Line 363: we added capitalization to the Figure 2.

Line 408 remove one dot

A: We removed it. Thank you.

Line 416 “Supplemental table 2“ change for „Supplemental Table 2” thgrough the whole text (see also lines 423, 426, etc.)

A: Line 403: we replaced capitalization for all lines.

Line 421 create new paragraph

A: We created indent on the line 409.

Line 469 “M. abscessus subsp. abscessus“ change for „Mycobacterium abscessus subsp. abscessus“

A: Line 457: we changed it to full name.

Line 491 create new paragraph

A: Line 477: This paragraph is continuation of the previous paragraph on the efflux pumps found in our study and there is no need to create indent.

Line 517 remove [45] (it is quoted twice in one paragraph)

A: We removed [45] now is ref #46.

Reviewer 2 Report

Within this manuscript, Gorzynski et al brought an attention to an important topic, the intrinsic resistance to antibiotics of Mycobacterium abscessus, the agent of severe infections in immunocompromised patients.  Using transposon mutagenesis, the authors analyzed 2000 mutants for modulated sensitivity to antibiotics, discovered 134 mutants out of which they focused on 16 clones lacking genes encoding cell wall-associated transport systems. Two mutants lacking the genes for MmpL exporter pumps displayed reduced survival in THP-1 macrophages treated with amikacin.

The study is conceptually correct and scientifically sound. Nevertheless, I have some major and minor comments that need to be addressed before the manuscript can be considered for acceptance.

Major comments:

  1. Paragraph 2.9. I have some doubts on differentiation and infection protocol. THP-1 monocytes were differentiated with 20 ng/ml of PMA, which corresponds roughly to 33 nM PMA. However, most of the PMA-driven differentiation protocols use 100 nM PMA. Furthermore, it is not clear how long the monocytes were treated with PMA, one or two days? In any case, two days of resting seems to be quite long, especially in combination with another 5 days of infection. After such a long period of time in the absence of PMA, the THP-1 cells lose the adherence (Spano et al. 2013, Lund et al. 2016). In our hands, we also see that substantial part of cells dedifferentiate back to monocytes (cells detach and get round). Were the THP-1 macrophages tested under microscope or analyzed by FACS? Were the THP-1 monolayers washed before lysis with Triton? These are very important issues.

Furthermore, the cells were incubated for 2 h with bacteria, how was the contact with macrophages achieved? And how was the sample 1 h post infection prepared considering that after 2 h of infection the cells were further treated with amikacin for additional 2 h.

Is it possible that high concentration of amikacin (200 ug/ml) cause some cytotoxic effect towards macrophages?

  1. Paragraph 2.11. What is the penetration kinetics of amikacin into macrophage cells? It seems that it is rather slow (4 days, Maurin and Raoult 2001), are the effects observed 1 and 3 days pi shown in Fig. 5B caused solely by amikacin?
  2. Figure 3. Did the authors tested higher MOI values?
  3. I am not sure whether the data in Figure 5 bring any additional knowledge when compared to results presented in Table 4. Both MAB_0937c and MAB_1137c mutants display higher sensitivity to amikacin compared to parental strain in liquid medium, and apparently, also within macrophages.
  4. Authors should at least attempt to explain why the mutants displaying reduced resistance to antimicrobials did not show any effect in macrophage model.

Minor comments

  1. Line 218. The inhibition of growth was analyzed only visually, did the authors tried for at least some samples to plate the cells to prove that none of the cells survived?
  2. While the manuscript is overall well written, there are several typos to be corrected such as:

Line 279: should be “heat-inactivated”

Legend to Figure 3, should be “20 ng/ml”

Line 420: “16S rRNA gene”

Line 457: replace “pimp” with “pump”

Line 612: “we could demonstrate”

at several places in Discussion the authors switch between “TetR” and “tetR” terms

Author Response

We would like to thank the reviewer for thorough review and suggestions that we incorporated in the paper. All changes are mark in red.

Major comments:

  1. Paragraph 2.9. I have some doubts on differentiation and infection protocol. THP-1 monocytes were differentiated with 20 ng/ml of PMA, which corresponds roughly to 33 nM PMA. However, most of the PMA-driven differentiation protocols use 100 nM PMA. Furthermore, it is not clear how long the monocytes were treated with PMA, one or two days? In any case, two days of resting seems to be quite long, especially in combination with another 5 days of infection. After such a long period of time in the absence of PMA, the THP-1 cells lose the adherence (Spano et al. 2013, Lund et al. 2016). In our hands, we also see that substantial part of cells dedifferentiate back to monocytes (cells detach and get round). Were the THP-1 macrophages tested under microscope or analyzed by FACS? Were the THP-1 monolayers washed before lysis with Triton? These are very important issues.

A: The PMA concentrations used for THP-cell differentiation ranges from 10 -100 ng/ml. Over decades of experiments with THP-1 monocytes in my lab, we found that to have confluent macrophage monolayers (without over activating macrophages) ranges in 20 -50 ng/ml, and 20 ng/ml concentration is optimal when using PMA purchased from the Sigma-Aldrich. If higher concentrations are used, macrophages tend to accumulate lots of lysosomes in the cytosol during differentiation and several days after. We term them as “angry macrophages”. To avoid this, we use 20 ng/ml overnight and then rest cells for additional 48 h at which time-point we infect cells (Lines 269-271).

In regards of the reviewer’s comment “In our hands, we also see that substantial part of cells dedifferentiate back to monocytes”. If macrophages are properly differentiated, they don’t reverse to monocytes, especially, when they are infected with bacteria. The proper differentiation of cells is significantly influenced how cells are maintained, THP-1 cell growth phase and the monocytes concentrations in the original flasks that are used for differentiation with PMA and seeding into the well-plates. These monolayers are good for 7 days.

We always observe THP-1 monolayers under the inverted microscope at each procedure, infection, washings, treatments and at each day of the experiment. Before cell lysis and after to check that they are fully lysed. First, the culture media is removed and then lysed. Why cells would be washed before lysis?

Furthermore, the cells were incubated for 2 h with bacteria, how was the contact with macrophages achieved? And how was the sample 1 h post infection prepared considering that after 2 h of infection the cells were further treated with amikacin for additional 2 h.

A: Bacteria are added directly on the monolayer, plates gently inverted and placed in the incubator for 2h.

We lysed cells at 2h as a baseline, and we made this correction to the 1h time-point. Line: 277. Thank you.

Is it possible that high concentration of amikacin (200 ug/ml) cause some cytotoxic effect towards macrophages?

A: No, cytotoxic affect at this concentration. For example, the concentration used to kill M. avium strain is 400 ug/ml and it is not toxic either.

  1. Paragraph 2.11. What is the penetration kinetics of amikacin into macrophage cells? It seems that it is rather slow (4 days, Maurin and Raoult 2001), are the effects observed 1 and 3 days pi shown in Fig. 5B caused solely by amikacin?

A: While it is slow, we are using the bactericidal concentration of AMK to treat cells. The killing of mycobacteria is very challenging. At low bacterial concentrations, it takes 5 days to completely clear bacteria in the tube where the host cell penetration is not a factor at all.

  1. Figure 3. Did the authors tested higher MOI values?

A: We used more clinically relevant bacterial concentration (1 bacteri:1cell). If we use, for example, 10 bacteria to 1 cell, since MAB is fast growing, it will overgrow cells and lyse cells within 3 days.

  1. I am not sure whether the data in Figure 5 bring any additional knowledge when compared to results presented in Table 4. Both MAB_0937c and MAB_1137c mutants display higher sensitivity to amikacin compared to parental strain in liquid medium, and apparently, also within macrophages.

A: The Figure 5 shows MAB time killing dynamics in cultured macrophages. The Table 4 lists MIC/BC concentrations in the bacterial growth media in vitro. 

  1. Authors should at least attempt to explain why the mutants displaying reduced resistance to antimicrobials did not show any effect in macrophage model.

A: The overall answer is that these genes most likely are not essential factors for intracellular MAB and thus we do not observe attenuation of MAB.

Minor comments

  1. Line 218. The inhibition of growth was analyzed only visually, did the authors tried for at least some samples to plate the cells to prove that none of the cells survived?

A: The MIC concentrations were visually determined, while BC concentration tubes were centrifuged and plated on the growth media to confirm there were no bacterial growth. We added this sentence on Lines: 210-212. Thank you.

  1. While the manuscript is overall well written, there are several typos to be corrected such as:

Line 279: should be “heat-inactivated”

A: We fixed to heat inactivated on line 268.

Legend to Figure 3, should be “20 ng/ml”

A: We changed to ng on the line 392.

Line 420: “16S rRNA gene”

A: We added rRNA line 407.

Line 457: replace “pimp” with “pump”

A: We fixed to pump on line 445.

Line 612: “we could demonstrate”

A: Line 603: we changed can to could.

at several places in Discussion the authors switch between “TetR” and “tetR” terms

A: We replaced all to TetR.

Round 2

Reviewer 2 Report

We would like to thank the reviewer for thorough review and suggestions that we incorporated in the paper. All changes are mark in red.

Major comments:

  1. Paragraph 2.9. I have some doubts on differentiation and infection protocol. THP-1 monocytes were differentiated with 20 ng/ml of PMA, which corresponds roughly to 33 nM PMA. However, most of the PMA-driven differentiation protocols use 100 nM PMA. Furthermore, it is not clear how long the monocytes were treated with PMA, one or two days? In any case, two days of resting seems to be quite long, especially in combination with another 5 days of infection. After such a long period of time in the absence of PMA, the THP-1 cells lose the adherence (Spano et al. 2013, Lund et al. 2016). In our hands, we also see that substantial part of cells dedifferentiate back to monocytes (cells detach and get round). Were the THP-1 macrophages tested under microscope or analyzed by FACS? Were the THP-1 monolayers washed before lysis with Triton? These are very important issues.

A: The PMA concentrations used for THP-cell differentiation ranges from 10 -100 ng/ml. Over decades of experiments with THP-1 monocytes in my lab, we found that to have confluent macrophage monolayers (without over activating macrophages) ranges in 20 -50 ng/ml, and 20 ng/ml concentration is optimal when using PMA purchased from the Sigma-Aldrich. If higher concentrations are used, macrophages tend to accumulate lots of lysosomes in the cytosol during differentiation and several days after. We term them as “angry macrophages”. To avoid this, we use 20 ng/ml overnight and then rest cells for additional 48 h at which time-point we infect cells (Lines 269-271).

In regards of the reviewer’s comment “In our hands, we also see that substantial part of cells dedifferentiate back to monocytes”. If macrophages are properly differentiated, they don’t reverse to monocytes, especially, when they are infected with bacteria.

R: this is not true, several human pathogens are capable to dedifferentiate macrophage cells to monocytes as a part of their survival strategy. That is why it is important to monitor the cells under microscope or perform FACS as to check important cytomarkers (CD11b, CD68, CD14).

The proper differentiation of cells is significantly influenced how cells are maintained, THP-1 cell growth phase and the monocytes concentrations in the original flasks that are used for differentiation with PMA and seeding into the well-plates. These monolayers are good for 7 days.

R: I am still confused how you can keep macrophages for such a long time attached to wells (in the absence of PMA).

We always observe THP-1 monolayers under the inverted microscope at each procedure, infection, washings, treatments and at each day of the experiment. Before cell lysis and after to check that they are fully lysed. First, the culture media is removed and then lysed. Why cells would be washed before lysis?

R: Since you do not keep AMK during infection (at least I did not find it in the protocol), all bacteria that escape from lysed or apoptotic macrophages will stay alive in the RPMI medium and will replicate. Without AMK treatment or thorough washing steps you will count these cells together with intracellular bacteria. This may bias your data. Did you try to plate culture medium before the macrophages were lysed to check for extracellular bacteria?

Furthermore, the cells were incubated for 2 h with bacteria, how was the contact with macrophages achieved? And how was the sample 1 h post infection prepared considering that after 2 h of infection the cells were further treated with amikacin for additional 2 h.

A: Bacteria are added directly on the monolayer, plates gently inverted and placed in the incubator for 2h.

R: We do work with another human pathogen, but in our experience we need to gently centrifuge the bacteria to get proper contact with macrophages.

We lysed cells at 2h as a baseline, and we made this correction to the 1h time-point. Line: 277. Thank you.

Is it possible that high concentration of amikacin (200 ug/ml) cause some cytotoxic effect towards macrophages?

A: No, cytotoxic affect at this concentration. For example, the concentration used to kill M. avium strain is 400 ug/ml and it is not toxic either.

  1. Paragraph 2.11. What is the penetration kinetics of amikacin into macrophage cells? It seems that it is rather slow (4 days, Maurin and Raoult 2001), are the effects observed 1 and 3 days pi shown in Fig. 5B caused solely by amikacin?

A: While it is slow, we are using the bactericidal concentration of AMK to treat cells. The killing of mycobacteria is very challenging. At low bacterial concentrations, it takes 5 days to completely clear bacteria in the tube where the host cell penetration is not a factor at all.

R: this is exactly my point, which you did not respond. You see reduced survival in both mutants already 24 h pi, so is not it possible that these pumps may export other toxic compounds in addition to AMK?

  1. Figure 3. Did the authors tested higher MOI values?

A: We used more clinically relevant bacterial concentration (1 bacteri:1cell). If we use, for example, 10 bacteria to 1 cell, since MAB is fast growing, it will overgrow cells and lyse cells within 3 days.

R: I understand, MOI 1 is more biologically relevant, but maybe at higher MOI values you could better distinguish mutants that replicate slower than wt. Maybe the mutants would lyse the cells later than within 3 days. Did you test this?

  1. I am not sure whether the data in Figure 5 bring any additional knowledge when compared to results presented in Table 4. Both MAB_0937c and MAB_1137c mutants display higher sensitivity to amikacin compared to parental strain in liquid medium, and apparently, also within macrophages.

A: The Figure 5 shows MAB time killing dynamics in cultured macrophages. The Table 4 lists MIC/BC concentrations in the bacterial growth media in vitro.

R: Does not respond my question, I am aware that these are two different experiments, but my point was that the outcome is the same, mutants are more sensitive to AMK than wt. One could at least discuss this observations.

  1. Authors should at least attempt to explain why the mutants displaying reduced resistance to antimicrobials did not show any effect in macrophage model.

A: The overall answer is that these genes most likely are not essential factors for intracellular MAB and thus we do not observe attenuation of MAB.

R: I meant that the authors should discuss this observation within the manuscript, not only with me. I do not see any changes in the Discussion part, which, somehow, is anyway quite succinct.

Author Response

Major comments:

  1. Paragraph 2.9. I have some doubts on differentiation and infection protocol. THP-1 monocytes were differentiated with 20 ng/ml of PMA, which corresponds roughly to 33 nM PMA. However, most of the PMA-driven differentiation protocols use 100 nM PMA. Furthermore, it is not clear how long the monocytes were treated with PMA, one or two days? In any case, two days of resting seems to be quite long, especially in combination with another 5 days of infection. After such a long period of time in the absence of PMA, the THP-1 cells lose the adherence (Spano et al. 2013, Lund et al. 2016). In our hands, we also see that substantial part of cells dedifferentiate back to monocytes (cells detach and get round). Were the THP-1 macrophages tested under microscope or analyzed by FACS? Were the THP-1 monolayers washed before lysis with Triton? These are very important issues.

A: The PMA concentrations used for THP-cell differentiation ranges from 10 -100 ng/ml. Over decades of experiments with THP-1 monocytes in my lab, we found that to have confluent macrophage monolayers (without over activating macrophages) ranges in 20 -50 ng/ml, and 20 ng/ml concentration is optimal when using PMA purchased from the Sigma-Aldrich. If higher concentrations are used, macrophages tend to accumulate lots of lysosomes in the cytosol during differentiation and several days after. We term them as “angry macrophages”. To avoid this, we use 20 ng/ml overnight and then rest cells for additional 48 h at which time-point we infect cells (Lines 269-271).

In regards of the reviewer’s comment “In our hands, we also see that substantial part of cells dedifferentiate back to monocytes”. If macrophages are properly differentiated, they don’t reverse to monocytes, especially, when they are infected with bacteria.

R: this is not true, several human pathogens are capable to dedifferentiate macrophage cells to monocytes as a part of their survival strategy. That is why it is important to monitor the cells under microscope or perform FACS as to check important cytomarkers (CD11b, CD68, CD14).

A: We do not observe macrophage monolayers to reverse back to monocytes in our assays.  Mycobacterial infections including M. tuberculosis, M. avium, M. abscessus or M. avium subsp. paratuberculosis stimulate immune responses (in the infected and uninfected cells within the same monolayer) that don’t result of macrophages to dedifferentiate to monocytes. As I mentioned in my previous response, we always monitor cells under the microscope. The reviewer would agree that initially when the differentiation protocol is established in the lab, through FACS, observations under the microscope and experiences gained working with these cells, you won’t perform FACS for markers mentioned above for every single infection assay that we do in the lab.

The proper differentiation of cells is significantly influenced how cells are maintained, THP-1 cell growth phase and the monocytes concentrations in the original flasks that are used for differentiation with PMA and seeding into the well-plates. These monolayers are good for 7 days.

R: I am still confused how you can keep macrophages for such a long time attached to wells (in the absence of PMA).

A: There are many published papers as well as discussions between scientists available online on range of concentrations and time exposure of PMA to THP-1 cells. PMA treatment in all published articles does not go beyond 48h (it’s either 24h or 48h). If you keep PMA longer, it is cytotoxic to cells. Also, if higher concentrations are used for 24h, macrophages get overactivated producing high levels of chemokines and cytokines that affect the experimental outcome due to the PMA. In the past, we also observed that when we used PMA from Fisher BioReagents vs Sigma-Aldrich exactly the same concentrations, the monocyte differentiation outcome was very different (PMA from Fisher BioReagents failed to stimulate cells the same way as Sigma). Since then, we only purchase this reagent from Sigma, store at 1mg stocks and always prepare fresh working solution when used in experiments.

We always observe THP-1 monolayers under the inverted microscope at each procedure, infection, washings, treatments and at each day of the experiment. Before cell lysis and after to check that they are fully lysed. First, the culture media is removed and then lysed. Why cells would be washed before lysis?

R: Since you do not keep AMK during infection (at least I did not find it in the protocol), all bacteria that escape from lysed or apoptotic macrophages will stay alive in the RPMI medium and will replicate. Without AMK treatment or thorough washing steps you will count these cells together with intracellular bacteria. This may bias your data. Did you try to plate culture medium before the macrophages were lysed to check for extracellular bacteria?

A: This experiment is performed to establish if MAB mutants are attenuated in the intracellular growth within macrophages due to the gene defects that was knocked out. We check integrity of cell monolayers every day and yes, we check supernatants (RPMI) at each time-point by plating them on the 7H10 agar plates to see if we have extracellular bacteria growing (we added lines 280-282 in red). To highlight, at the MOI of 1 bacterium to 1 cell, bacteria do not cause cell lysis or apoptosis. Generally, we observe intracellular MAB exit from cells and, thus, extracellular bacterial growth when MOI is 5 bacteria (or higher): 1 cell. In this case majority of macrophages are detaching at day 3 and under the microscope it is apparent the growth of extracellular bacteria. This is why it is also important to use lower MOIs for MAB infection to capture any observations initiated by intracellular bacteria.

Furthermore, the cells were incubated for 2 h with bacteria, how was the contact with macrophages achieved? And how was the sample 1 h post infection prepared considering that after 2 h of infection the cells were further treated with amikacin for additional 2 h.

A: Bacteria are added directly on the monolayer, plates gently inverted and placed in the incubator for 2h.

R: We do work with another human pathogen, but in our experience we need to gently centrifuge the bacteria to get proper contact with macrophages.

A: Yes, I agree with reviewer, and we also synchronize via 10 min 250-300 rpm centrifugation if we incubate bacteria with cells for 1 hour or less time. But when we use 2-hour incubation/infection, this is enough time for all bacteria to have contact with cells.

We lysed cells at 2h as a baseline, and we made this correction to the 1h time-point. Line: 277. Thank you.

Is it possible that high concentration of amikacin (200 ug/ml) cause some cytotoxic effect towards macrophages?

A: No, cytotoxic affect at this concentration. For example, the concentration used to kill M. avium strain is 400 ug/ml and it is not toxic either.

  1. Paragraph 2.11. What is the penetration kinetics of amikacin into macrophage cells? It seems that it is rather slow (4 days, Maurin and Raoult 2001), are the effects observed 1 and 3 days pi shown in Fig. 5B caused solely by amikacin?

A: While it is slow, we are using the bactericidal concentration of AMK to treat cells. The killing of mycobacteria is very challenging. At low bacterial concentrations, it takes 5 days to completely clear bacteria in the tube where the host cell penetration is not a factor at all.

R: this is exactly my point, which you did not respond. You see reduced survival in both mutants already 24 h pi, so is not it possible that these pumps may export other toxic compounds in addition to AMK?

A: Yes, it is possible that these pumps not only transport drugs but are involved in export of other factors. If MAB exports toxic compounds through these pumps, then I would except the gene knockout mutants to display some growth defect (reduced number vs parental strain) in vitro as well. Both mutants grow in the culture the same way as the parental strain. 

What we can demonstrate in this experiment (Figure 5B) is that, when the efflux pumps are knockout out, we see attenuation in MAB intracellular tolerance against AMK and improved AMK treatment outcome, and complementation confirms that it is directly associated with selected pumps.  

Figure 3. Did the authors tested higher MOI values?

A: We used more clinically relevant bacterial concentration (1 bacteri:1cell). If we use, for example, 10 bacteria to 1 cell, since MAB is fast growing, it will overgrow cells and lyse cells within 3 days.

R: I understand, MOI 1 is more biologically relevant, but maybe at higher MOI values you could better distinguish mutants that replicate slower than wt. Maybe the mutants would lyse the cells later than within 3 days. Did you test this?

A: No, we did not test MAB infection at higher MOI.

  1. I am not sure whether the data in Figure 5 bring any additional knowledge when compared to results presented in Table 4. Both MAB_0937c and MAB_1137c mutants display higher sensitivity to amikacin compared to parental strain in liquid medium, and apparently, also within macrophages.

A: The Figure 5 shows MAB time killing dynamics in cultured macrophages. The Table 4 lists MIC/BC concentrations in the bacterial growth media in vitro.

R: Does not respond my question, I am aware that these are two different experiments, but my point was that the outcome is the same, mutants are more sensitive to AMK than wt. One could at least discuss this observations.

A: Because in the intracellular environment mycobacteria goes through metabolic remodeling and phenotypic changes that differs from extracellular state within the growth media, it is important to validate all observations that we see in vitro at least in the tissue culture system. In the discussion we highlight why we did this experiment; Lines 596-609 in red.

  1. Authors should at least attempt to explain why the mutants displaying reduced resistance to antimicrobials did not show any effect in macrophage model.

A: The overall answer is that these genes most likely are not essential factors for intracellular MAB and thus we do not observe attenuation of MAB.

R: I meant that the authors should discuss this observation within the manuscript, not only with me. I do not see any changes in the Discussion part, which, somehow, is anyway quite succinct.

A: We have many examples in science that some genes of bacterial pathogenicity that do not display the deficiency in cell culture model but they are essential for survival in vivo. We tested 16 different transport related mutants (of reduced resistance to antibiotic) in the macrophage infection assay; all of them are involved in different substrate and molecule transport and possibly many other functions that have not been characterized yet. To discuss why these different transport mutants do not show defect in survival within macrophages (or grow the same way as parental strain) will be all speculation. The goal of our project was to discover MAB factors that are associated with antibiotic tolerance, and then may be also involved in intracellular growth/survival as well.